# Fabrication and Performance Evaluation of Gelatin/Sodium Alginate Hydrogel-Based Macrophage and MSC Cell-Encapsulated Paracrine System with Potential Application in Wound Healing

**DOI:** 10.3390/ijms24021240

**Published:** 2023-01-08

**Authors:** Hang Yao, Xiaohui Yuan, Zhonglian Wu, Sumin Park, Wang Zhang, Hui Chong, Liwei Lin, Yuanzhe Piao

**Affiliations:** 1School of Chemistry and Chemical Engineering, Yangzhou University, Yangzhou 225009, China; 2Department of Applied Bioengineering, Graduate School of Convergence Science and Technology, Seoul National University, Seoul 08826, Republic of Korea; 3Advanced Institutes of Convergence Technology, Suwon 16229, Republic of Korea

**Keywords:** wound healing dressing, paracrine system, biocompatibility, 3D cell culture, natural polymer

## Abstract

A gelatin/sodium alginate-based hydrogel microsphere has been fabricated after reaction condition optimization. Macrophages (RAW246.7) and adipose mesenchymal stem cells (ADSC) have been subsequently encapsulated in the microsphere in order to construct a 3D paracrine system for wound healing treatment. The synthesized microsphere displayed neglectable cytotoxicity toward both encapsulated cells until 10 days of incubation, indicating promising biocompatibility of the microsphere. A qRT-PCR and ELISA experiment revealed positive regulation of cytokines (Arg-1, IL-6, IL-8, IL-10, bFGF, HGF, VEGF, TLR-1, and CXCL13) expression regarding macrophage phenotype transformation and anti-inflammatory performance both inside the microsphere and in the microenvironment of established in vitro inflammatory model. Additionally, positive tendency of cytokine expression benefit wound healing was more pronounced in a fabricated 3D paracrine system than that of a 2D paracrine system. Furthermore, the 3D paracrine system exhibited more efficiently in the wound healing rate compared to the 2D paracrine system in an in vitro model. These results suggested the current paracrine system could be potentially used as a robust wound healing dressing.

## 1. Introduction

Biobased and synthetic dressing material that could build a barrier against pathogen and refrain from heat loss and mimic a wound healing environment were widely used in wound healing [1,2]. Conventional synthetic dressing material was generally feathered with low cost but displayed the drawbacks of poor antibacterial performance together with a limited effect on accelerating wound healing [3]. Alternatively, biomaterials, such as collagen, chitosan, and hyaluronic acid exhibited better biocompatibility and could accelerate wound healing by decoration with growth factors [4,5]. For instance, fibroblast growth factor (bFGF) was complexed with negatively charged heparin, and the resulting hybrid material displayed promising performance in wound healing therapy [6]. Loading bFGF into hydrogels was proved to be another efficient approach for combating wound healing, in the aspect of enhanced dermis and granulation tissue growth and blood vessel formation [7]. Yet, this strategy displayed drawbacks of limited regulation capability of growth factor and denaturation of bioactive components [8,9,10]. Consequently, novel therapeutic systems were still on high demand.

The paracrine effect referred to the cellular behavior regulation by cytokines secreted from neighboring “working cells” [11,12]. Typically, paracrine cells included endocrine cells, mesenchymal stem cells (MSCs), macrophages, and islet D cells [13]. In details, macrophages displayed the functions of debridement, formation of blood vessels and matrix synthesis [14]. Macrophages can be classified as M1 and M2 phenotypes. The M1 phenotype could activate inflammatory reaction, which played a critical role in anti-inflammation in the early stage of inflammation. M2 phenotype was in charge of wound tissue reconstruction and reducing the side effect of an over-anti-inflammation reaction caused by M1 macrophages [14]. Thus, the balance of M1 and M2 was critical in the process of wound healing [15]. In addition to macrophages, MSCs as paracrine cells received intense attention in recent years [16,17]. MSCs showed the antibacterial function of by secreting antibacterial peptide (LL-37) and interleukin-6 (IL-6) that could reduce apoptosis of neutrophils caused by external stimulations [18]. Additionally, MSCs could secrete and transform M1 to M2 macrophages. Therefore, a combination of macrophages and MSCs further improve the wound healing outcome of the paracrine system.

Current wound healing by the paracrine system was mainly achieved by direct cell injection method. This approach would arise with the problems of limited in vivo colonization rate and relative tissue repairing capability, long in vivo migration, and differentiation rate and risk of immune exclusion reaction [19,20,21]. Consequently, addressing these issues are currently main hurdles for the paracrine system. A 3D hydrogel cell medium allowed more intercellular and cell–matrix interactions in contrast to a 2D cell culture and was closer to a physiological condition [22,23,24,25]. It was reported that metabolic level, cellular stress response, cellular signal transduction together with the protein transportation level in 3D cell culture was higher than that of in a 2D cell culture [26]. Thus, the paracrine system built on 3D cell culture might be a potential alternative for wound healing.

Gelatin and sodium alginate were characterized with promising biocompatibility and widely applied in the fabrication of biomaterials, including wound healing dressing [27,28,29]. Gelatin could facilitate cell adhesion and proliferation and reduce the risks of potential immune reactions [30]. However, gelatin displayed comprised mechanical properties and was unsuitable to fabricate 3D cell culture without chemical modification [31]. However, toxic chemical reagents were frequently involved in such a modification process [32]. Sodium alginate displayed promising mechanical properties and was widely used as a component for 3D hydrogel cell culture medium [33,34,35,36]. Upon mixing the two natural biopolymers and finely tuning the ratio, a robust paracrine system targeting wound healing might be established, taking merits of both polymers. Furthermore, the synergistic paracrine system that contained more than one functional cell was rarely used in the field of wound repair. In order to test the feasibility of this bifunctional cell loading strategy. We fabricated a corresponding paracrine system for wound healing by encapsulation of macrophages (RAW246.7) and adipose mesenchymal stem cells (ADSCs) in gelatin–sodium alginate-based hydrogel microsphere in this manuscript. The resulting system exhibited promising biocompatibility and indeed accelerated the wound healing rate in an in vitro model. In-depth investigations revealed M1 and M2 macrophages were well balanced in this paracrine system. Moreover, anti-inflammatory and pro-wound healing cytokines secreted by macrophages and ADSC cells were observed to be upregulated inside the paracrine and the wound environment compared to that of a control 2D paracrine system.

## 2. Results

### 2.1. Influences of Reaction Conditions on Microsphere Yield and Morphology

Three factors were evaluated on the yield and morphology of microspheres, including the reagent concentration in water, water–oil ratio, and stirring rate. The corresponding influences of the above-mentioned factors on the microsphere is shown in Figure 1a. We first fixed the water–oil ratio at 1:2.5 and a stirring rate at 500–600 rpm and tuned the concentration of gelatin and sodium alginate in water. Under this condition, increasing gelatin and sodium alginate concentration resulted in a slight reduction in the microsphere yield. In details, the yield was 97%, 89%, and 86% under the concentration of 1% (*w/v*), 1.5% (*w/v*), and 2% (*w/v*), respectively. The average diameter and distribution of the microsphere prepared in gelatin/sodium alginate concentration of 1% (*w/v*), 1.5% (*w/v*), and 2% (*w/v*) was 370, 420, and 560 μm and 150–650, 200–800, and 200–1500 μm, respectively (Figure 1b–d). The morphology of the microsphere in all three conditions resembled smooth drops (Appendix A). A wide diameter distribution might not facilitate promising encapsulation of a cell in a microsphere. A higher reagent concentration enabled better crossing efficiency and stability of the microsphere. Thus, concertation of 1.5% (*w/v*) was fixed and the water–oil ratio and stirring rate was further evaluated for optimization of microsphere preparation. Tuning the water–oil ratio did not obviously change the morphology of the microsphere (Appendix A). The yield of the microsphere in water–oil ratio of 1:2, 1:2.5, and 1:3 was 78%, 88%, and 83%, respectively (Figure 1e). Changing the water–oil ratio to 1:3 in the current condition caused an increase in the average microsphere diameter to 480 μm with distribution of 100–1100 μm. When the water–oil ratio was tuned to 1:2, the average diameter amounted to 360 μm with a distribution of 150–600 μm (Figure 1f–h). Therefore, the water–oil ratio of 1:2.5 was used as an optimized condition. Next, we evaluated the impacts of the stirring rate on the diameter and distribution. Slowing the stirring rate to 450–550 rpm resulted in an increase in the diameter to ~880 μm with a distribution of 200–2000 μm. Increasing the stirring rate to 500–600 rpm yielded a smaller average diameter of 420 μm with a narrower distribution of 200–800 μm. Further increasing the stirring rate to 550–650 yielded a bigger average diameter of 590 μm with a wider distribution of 200–1800 μm. Thus, stirring rate of 500–600 rpm was chosen as the optimized condition for microsphere preparation.

### 2.2. Influence of Cell Concentration on Encapsulation Rate in Microsphere

The encapsulation of cells into a microsphere is described in Section 4.4. As shown in Figure 2a,b, the cell encapsulation rate was dependent on the cell density, especially for RAW264.7 cells. Upon the density of a HSF cell amounted to 0.50 × 10^7^/mL, the cell encapsulation rate was determined to be ~13% using a total DNA concentration as reference. The number of encapsulated HSF cells was correspondingly determined to be 2.00 × 10^6^/g. When RAW264.7 cell was encapsulated at the density of 0.50 × 10^7^/mL, the encapsulation rate was calculated to be ~40%. The encapsulated cell number was 5.00 × 10^6^/g. Increasing the density to 1.00 and 1.50 × 10^7^/mL, the encapsulated rate was determined to be ~32% and ~28, respectively. The corresponding encapsulated cell number was determined to be 6.50 and 7.80 × 10^6^/g.

### 2.3. Influence of Cell Encapsulation on Microsphere Size and Morphology

A (RAW264.7 and HSF) cell-loaded microsphere was prepared by addition cells in the optimized microsphere preparation process. As shown in Figure 3, the encapsulation of cells caused reduction of diameter together with narrowing of size distribution. In details, an HSF-encapsulated (density of 0.50 × 10^7^/mL) microsphere displayed an average diameter of 145 μm with a distribution range of 30–220 μm (Figure 3a). Microsphere-encapsulated RAW246.7 cells displayed a slight cell density dependent on diameter and distribution. The diameter was 130 (cell density of 1.00 × 10^7^/mL) and 135 μm (cell density of 0.50 and 1.50 × 10^7^/mL). The distribution of the three microspheres centered in the range of 30–280 μm (Figure 3b–d).

### 2.4. Biocompatibility of Microsphere

The microsphere provides a 3D environment that provides channels of oxygen and nutrients intake and metabolism delivery [37]. Thus, we evaluated the biocompatibility of the microsphere. As shown in Figure 4, the microsphere did not display an obvious toxicity toward HSF encapsulated at the density of 0.50 × 10^7^/mL within 10 days of incubation time. RAW246.7 cell was observed to slightly proliferate in the microsphere at a density of 0.50, 1.00, and 1.50 × 10^7^/mL in day 1, 4, and 7. On day 10, a slight reduction in O. D. at 450 nm was observed, indicating a tiny amount of toxicity toward the RAW246.7 cell.

### 2.5. Influence on Expression of Macrophage 2 Phenotype and Anti-Inflammatory Genes in Encapsulated RAW246.7 and ADSC Cells

It was well reported that RAW246.7 cells can be polarized and transferred to macrophage 2 phenotype by cytokines, which could promote wound healing in a 2D environment [38]. In addition, ADSC cells were reported to secrete anti-inflammatory proteins that could also assist wound healing [19]. Thus, we tested the corresponding anti-inflammatory gene expression of the two cells in a microsphere. *CD206*, *Arg-1* (Arginase-1) and *IL-10* (Interleukin-10) were selected as classical markers for anti-inflammatory behavior. In details, *Arg-1* was responsible for promoting collagen synthesis and cell proliferation [39]. *IL-10* displayed an inhibition of the activation, migration, and adhesion of inflammatory cells [40]. These two markers were highly expressed in M2 macrophages [41]. As shown in Figure 5, all the genes were over-expressed in encapsulated polarized RAW246.7 cells. Noticeably, the expression level of *Arg-1* in the microsphere enhanced over 100 folds against that expressed in the 2D environment (*p* < 0.01). The expression level of the Arg-1 gene also enhanced over 1000 folds compared to non-polarized RAW246.7 cells (*p* < 0.0001). The expression level of CD206 of polarized RAW246.7 cells was 100 times higher compared to that non-polarized RAW246.7 cells in the microsphere (*p* < 0.01). Interestingly, non-polarized RAW246.7 cells displayed a comparable CD206 expression level compared to that of polarized cells in the microsphere (*p* < 0.01). *bFGF*, *HGF*, *VEGF*, *IL-6*, and *IL-8* were classical markers of ADSC cells. Encapsulation in the microsphere also caused enhanced expression of these genes. The relative fold change for bFGF, HGF, VEGF, IL-6, and IL-8 was 30 (*p* < 0.01), 2 (*p* < 0.05), 7 (*p* < 0.05), 80 (*p* <0.0001), and 28 (*p* < 0.001).

### 2.6. Influence on Expression of Inflammation Regulating Cytokines in Encapsulated RAW246.7 and ADSC Cells

Regulating the secretion of cytokines is one of the determining issues of the paracrine effect that could tune the anti-inflammatory progress. We selected classic pro-inflammatory factor (IL-6) and anti-inflammatory factors (IL-10) as a reference for an anti-inflammatory regulation evaluation. As shown in Figure 6a, all the experimental groups expressed a certain level of IL-6 (~75–80 pg). The 2D and M2 RAW246.7-encapsulated microsphere experimental group showed an enhanced IL-6 expression level (~35–50 pg, total amount of day 1 to 3) compared to the control (25 pg), but the ADSC- and ADSC/M2-encapsulated group showed reduced IL-6 expression that was comparable to that of control group (Figure 6b). All groups showed an obvious reduced IL-6 concentration after day 0, and the 3D functional cell-encapsulated groups were feathered with a mostly low IL-6 concentration. An expression level of IL-10 was analyzed in a similar manner compared to that of IL-10. As depicted in Figure 6d, the 3D experimental group showed relatively high expression of IL-10 compared to 2D group. RAW246.7 cells in the 2D environment caused the highest amount of IL-10 (~200 pg, Figure 6e). Generally, the 3D groups (RAW246.7, ADSC, and both cells) yielded a stable higher IL-10 concentration in day 0–3 (Figure 6f).

### 2.7. Influences of the Paracrine System on an Inflammatory Cell Phenotype in an In Vitro Inflammatory Regulation Model

The phenotypes of representative inflammatory M1 macrophage cell were analyzed in the fabricated paracrine system. Several pro-inflammatory markers, including *TNF- α*, *IL-1β*, *iNOS*, *CD86*, *CXCL9*, and *CCL5* were selected. As shown in Figure 7a–f, a slight upregulation of *TNF-α* (~4 folds) was observed in the 3D experimental groups compared to the control group. The 2D RAW246.7 cells condition resulted in an upregulation of *IL-1β* (~over 10 folds) and *CSCL9* (~100 folds, *p* < 0.05), and these genes were slightly upregulated to the 3D groups (~2 folds). The expression of rest genes was insignificantly influenced. Furthermore, the markers of the M1 to M2 macrophage transformation was detected. The 3D experimental group caused a significant upregulation of several corresponding gene markers compared to that of the 2D experimental groups. In details, *Arg-1* was upregulated by ~10 folds (3D ADSC over 2D ADSC, *p* < 0.01); *CD206* and *CSCL13* were upregulated by over 1000 folds (3D RAW246.7 over 2D RAW 246.7, *p* < 0.01, Figure 7g–i).

### 2.8. Effects of a Paracrine Entity on Cell Migration in an In Vitro Epithelial Recovery Model

The influence of a paracrine entity on epithelial recovery was evaluated based on the in vitro wound healing model in 2D and 3D environments. Figure 8a depicted the wound healing condition using the 2D paracrine entity. The wound healing percentage increased with incubation time. The RAW246.7 M2 group showed the most pronounced wound healing; the percentage was 30.72%, 46.74%, 60.96%, and 79.59% in 7, 14, 21, and 28 h. The wound healing percentage in ADSC group was 23.40%, 36.85%, 57.17%, and 66.81% in corresponding time. Additionally, the control group showed the slowest wound healing; the corresponding percentage was 19.93%, 34.72%, 46.96%, and 64.94%, respectively (Figure 8a,b). The 3D paracrine entity resulted in much faster wound healing. Noticeably, the 3D RAW246.7 M2 group resulted in 100% wound recovery in 21 h. Upon incubation of 28 h, all 3D experimental groups yielded 100% wound healing (Figure 8c,d).

## 3. Discussion

An over-inflammatory reaction is one of the main hurdles in wound healing. Correspondingly, benign regulation of an inflammatory reaction would facilitate quick access of a proliferation period in the process of wound healing. It was reported that the paracrine effect was one of the robust approaches to achieve inflammatory regulation [13]. In this study, we fabricated a paracrine cell dress and tested the effect on wound healing and conducted the analysis of the expression of a relevant inflammatory biomarker.

The paracrine cell-loaded wound healing dress was fabricated using bio-friendly gelatin and sodium alginate by a mono-emulsification method. Reaction conditions, including reagent concentration in water, water–oil ratio, and stirring rate, were found to play a significant role in the yield, size, and morphology of the desired microsphere. After screening, the optimized condition for a microsphere was a water–oil ratio of 1:2.5, a reagent concentration in water of 1.5% (*w/v*), and 500/600 rpm. This condition allowed the yield of 89%, an average diameter of 420 μm with distribution of 200–800 μm, and a relatively rough surface, which partially enabled promising cell loading and critical cytokines secretion. Cell (RAW264.7) encapsulation shrank the microsphere diameter size to around 130 μm (cell density of 1.00 × 10^7^/mL). This indicated more intense interactions between the cell surface and negative-charged molecular building blocks. Survival rate of encapsulated RAW246.7 cells at density of 0.50 × 10^7^/mL remained 100% until 10 days of incubation. The corresponding value dropped to around 90% on the 10th day when the cell density reached 1.00 × 10^7^/mL. A further increase in cell density to 1.50 × 10^7^/mL caused a slight reduction in cell survival rate, indicating a relatively small survival area of cells. Collectively, the paracrine cell encapsulated microspheres were feathered with suitable sizes that might promise sufficient necessary cytokine secretion in the process of wound healing and acceptable biocompatibility.

M2 phenotype of microsphere cells (RAW246.7) that were induced by polarization of M1 phenotype displayed the function of secreting anti-inflammatory factors. In contrast to the 2D cell culture method, the critical mRNA markers for the M2 phenotype were observed to be significantly upregulated (Arg-1, CD206 and IL-10) in the presence of a stimulator. Arg-1 could promote collagen synthesis and epithelial cells proliferation. IL-10 could inhibit the activation, migration, and adhesion of pro-inflammatory cells. Noticeably, IL-10 was even upregulated without inducement in a 3D environment, probably suggesting the 3D environment itself could enable the transformation of M2 macrophages. Stimulation of ADSC cells by pro-inflammatory factors and biomaterials was regarded as one of the efficient approaches to enhance the outcome of wound healing. Therefore, we examined the expression level of several biomarkers from ADSC cells. bFGF, HGF, VEGF, IL-6, and IL-8 could accelerate wound healing via different pathways [42]. In details, bFGF, HGF, and VEGF could promote blood vessel formation in wound tissue. IL-6 could influence the metabolism of the wound tissue [19]. IL-8 was responsible for neutrophil chemotaxis [17]. These genes were also significantly upregulated by encapsulated ADSC cells compared to the control group. In general, the addition of gelatin in the current material might provide function of assisting adhesion and migration of encapsulated cells and facilitate exerting paracrine system function. Furthermore, upregulation of biomarkers from both M2 phenotype macrophage and ADSC cells suggested the potential application as a wound healing dressing material of the fabricated microsphere.

Despite a successful expression of mRNA biomarkers that favored wound healing from the encapsulated cells. It remained important to test the capability of the paracrine system on regulating the expression level of key biomarkers that spread into an inflammatory environment. The protein expression level of classical pro-inflammatory (IL-6) and anti-inflammatory (IL-10) factors was evaluated in the supernatant of the established paracrine system in a different incubation time span. Initially (day 0), relatively high expression of IL-6 was observed in all experimental groups, suggesting successful transformation of M1 phenotype macrophage. The total amount of IL-6 from the supernatant of ADSC- and ADSC/RAW246.7-encapsulated microsphere was comparable to that of the control group and lower than that of the 2D paracrine system. This might indicate better anti-inflammatory capability of the 3D paracrine system. A time-dependent IL-6 concentration revealed that the 3D paracrine system could inhibit expression of this pro-inflammatory factor in a relatively long term (3 days). Contrary to the condition of pro-inflammatory marker IL-6, the 3D paracrine system caused a relatively enhanced accumulation of anti-inflammatory factor IL-10 in inflammatory environment. Despite RAW246.7 cells resulting in a high expression of IL-10, it remained a challenge to use 2D cells as efficient treatment of wound healing.

On top of the protein inflammatory biomarkers, we further evaluated the impact of 3D paracrine system on genetic biomarker expression in an inflammatory environment. Several pro-inflammatory mRNA biomarkers, including *TNF-α*, *IL-1β*, *iNOS*, *CD86*, *CXCL9*, and *CCL5*, were selected. Typically, the occurrence of a cytokine storm upon acute inflammation could cause pronounced upregulation of these mRNA biomarkers. The fabricated 3D paracrine system did not result in such a highly upregulation of these biomarkers as suggested the neglectable possibility of pro-inflammation occurrence. To note, the 3D paracrine system resulted in a slight downregulation of the inflammatory tissue-repairing gene CXCL9 compared to the 2D paracrine system. This might suggest the anti-inflammatory effect of the 3D paracrine system. Furthermore, an expression of classic anti-inflammatory mRNA biomarkers, including Arg-1, CD206, TLR-1, and CXCL13 was upregulated in an inflammatory environment in the presence of the 3D paracrine system compared to that of the 2D paracrine system. More importantly, the 3D paracrine system significantly accelerated the migration rate of human epithelium HSF cells compared to the 2D paracrine system. In conclusion, the 3D gel-phase paracrine system was highly possible of not triggering obvious pro-inflammatory reactions but rather switching on anti-inflammatory pathways that could facilitate wound tissue repairing.

## 4. Materials and Methods

### 4.1. Materials and Instruments

Chemicals for the microsphere preparation, including gelatin and sodium alginate, were purchased was purchased from Shanghai Aladdin Biochemical Technology Co., Ltd. (Shanghai, China). Regents for cell culture, qPCR, and ELISA were purchased from Beyond Time Co., Ltd. (Shanghai, China). Transwell kit was purchased from Thermo Fisher (USA). Primers for qRT-PCR were ordered from Beyond Time Co., Ltd. (Shanghai, China). Mouse RAW264.7, rabbit ADSC, and human HSF cells were purchased from Gibco (Grand Island, NY, USA). Fluorescent imaging and bright field photos were taken on IX83 fluorescent microscopy, Olympus (Hamburg, German). qRT-PCR was conducted on MyGo-Pro system, IT-IS (London, UK). Lyophilization was conducted on Christ-Alpha 1-2LD plus lyophilizer (Berlin, German). TEM images were taken on HT7800, ITACHI (Tokyo, Japan). Particle diameter analysis was conducted on ZEN360 ζ potential analyzer (London, UK).

### 4.2. Preparation of Gelatin/Sodium Alginate Microsphere

A total of 8.50 g of gelatin and 0.75 g of sodium alginate was dispersed in 50 mL of ddH_2_O, and the resulting mixture was heated at 60 °C for 2 h in order to obtain a uniform solution. A total of 20 mL of the solution was mixed with 50 mL of cooking oil and stirred for 4 min at room temperature and 4 °C for 10 min with 500 rpm. Then, 4 mL of CaCl_2_ solution (102 mM) was added, and the mixture was further stirred at 600 rpm for 4 min. Finally, the microsphere was washed with 1 × PBS (20 mL) for three times and lyophilized.

### 4.3. Cell Culture

Mouse RAW 264.7 cells were incubated in DMEM medium (containing 10% fetal bovine serum, 1 mM sodium pyruvate, 1% penicillin and streptomycin, and other essential components) at 37 °C in a 5% CO_2_ atmosphere environment. The cell was harvested by a centrifuge (240 rpm, 3 min) after treatment with trypsin at 37 °C for 5 min.

ADSC cells were incubated in DMEM/F12 medium (containing 10% fetal bovine serum, 1 mM ascorbic acid, 1% penicillin and streptomycin, 1% bFGF/F-12 growth factor, and other essential components) at 37 °C in a 5% CO_2_ atmosphere environment. The cell was harvested by a centrifuge (240 rpm, 3 min) after treatment with trypsin at 37 °C for 5 min.

HSF cells were incubated in DMEM medium (containing 15% fetal bovine serum, 1% penicillin and streptomycin, and other essential components) at 37 °C in a 5% CO_2_ atmosphere environment. The cell was harvested by a centrifuge (240 rpm, 3 min) after treatment with trypsin at 37 °C for 5 min.

### 4.4. Preparation of Cell Loaded Microsphere

A certain volume of as-prepared bacteria-free solution of gelatin/sodium alginate solution was mixed with 6 × 10^7^ RAW264.7 cells to make the final cell concentration, which amounted to 0.5 × 10^7^, 1.0 × 10^7^ and 1.5 × 10^7^/mL. The resulting mixture was next gently mixed with 20 mL of cooking oil followed by stirring at 500 rpm at room temperature for 4 min, stirring at 500 rpm at 4 °C for 10 min, and stirring at 600 rpm at room temperature for 4 min with the addition of 4 mL CaCl_2_ solution (102 mM). Subsequently, the microsphere was thoroughly washed with serum-free DMEM (10 mL) for 5 times and was incubated at 37 °C in a 5% CO_2_ atmosphere environment.

### 4.5. Determination of RAW264.7 Loading Ratio

A certain cell-loaded microsphere was added to 1mL of papain and was gently shaken. The cells were subsequently lysed overnight at 60 °C. The total DNA was then extracted followed by the standard protocol [43]. The total DNA weight was determined using a standard calf thyme DNA solution as reference. The total cell number was calculated using the formula:$$N_{cell}=W_{DNA}/7.70$$
where *N_cell_* denoted the total cell number in a microsphere and *W_DNA_* stood for the total DNA weight of the cell lysis solution. The corresponding cell-loading ratio was calculated by dividing *N_cell_* by the cell number in the microsphere preparation process.

### 4.6. Cytotoxicity of Microsphere

A cell-loaded microsphere was first incubated at 37 °C in a 5% CO_2_ atmosphere environment for 24 h. Then, the microsphere was re-dispersed with the addition of sodium alginate and transferred to a 24-well plate. CaCl_2_ solution was added into the 24-well plate to form a gel block, and the plate was incubated under the above condition for 10 days. On day 1, 4, 7, and 10, the liquid in the individual wells was removed and replaced with 550 μL culture medium and 55 μL of CCK working solution followed by incubation for 4 h. Then, 100 μL of supernatant liquid from an individual well was transferred into a 96-well plate and the O. D. value was measured by a plate reader to yield the cell cytotoxicity of microsphere.

### 4.7. Polarization Inducement of RAW264.7 Cells

The cells (not loaded) that were incubated in DMEM cell culture for 24 h were lysed and harvested by a centrifuge (240 rpm, 5 min). A total of 2 mL of cell dispersion (density of 1.0 ×10^5^/mL) was then seeded in a 6-well plate and incubated. A total of 90% of the well was covered by cells, and the culture medium was removed and replaced by a serum-free medium added to lipopolysaccharide (LPS, 100 ng/mL) and interferin-γ (INF-γ, 40 ng/mL). After incubation for 24 h, the cells would be transformed into a M1 phenotype. Further, M1 to M2 transformation would be achieved by incubation in a serum-free medium added to interleukin-4 (IL-4, 20 ng/mL) and interleukin-13 (IL-13, 20 ng/mL) for another 24 h. The cells were then incubated for another 24 h and harvested. Cells incubated without IL-4 and IL-13 were used as control.

Cell-loaded microspheres were incubated in DMEM cell culture for 24 h and collected by a cell filter and centrifuge. Microspheres with equal weight were then incubated in a serum-free DMEM cell culture with the addition of IL-4 (20 ng/mL) and IL-13 (20 ng/mL) in a 6-well plate for 24 h. Microspheres incubated without IL-4 and IL-13 were used as control.

### 4.8. Establishment of Inflammatory Regulation in vitro Model

A 2D model was established by seeding RAW264.7 cells (M1 phenotype, 2 × 10^5^/mL) in the bottom layer of a Transwell kit. Upon the cell coverage rate amounting to 90%, the culture medium was replaced by 2 mL of serum-free DMEM added to LPS (100 ng/mL) and INF-γ (40 ng/mL) and marked as Day1. The cells were incubated for 24 h (marked Day0) and half of the supernatant was collected followed by the addition of an equal volume of fresh culture medium. The M2 phenotype of RAW264.7 and ADSC cells were seeded in the upper layer of a Transwell kit and incubated at 37 °C in a 5% CO_2_ atmosphere environment for 24 h (marked Day1). The supernatant was collected and stored on Day1, Day2 and Day3 followed by an identical protocol (without the addition of RAW264.7 and ADSC cells after Day1). The samples of Day0, Day1, Day2, and Day3 were then subjected to an ELISA test for inflammatory regulation protein expression by standard protocol (Avidin-HRP-TMB method). The sample on Day3 was subjected to qRT-PCR test for inflammatory regulation mRNA expression.

The 3D model was established through resembling the procedure by using a cell-loaded microsphere instead of bare cells. The samples of Day0, Day1, Day2, and Day3 were then subjected to an ELISA test for inflammatory regulation protein expression analysis by standard protocol (Avidin-HRP-TMB method). The sample on Day3 was subjected to a qRT-PCR test for inflammatory regulation mRNA expression.

### 4.9. Establishment of Inflammatory Regulation in vitro Model

HSF cells (concentration of 2 × 10^5^/mL) were seeded in the bottom layer of a Transwell kit and incubated at 37 °C in a 5% CO_2_ atmosphere environment. When the cell covering ratio amounted to ~90%, the wound model was made using a cell scraper. Then, the wound was washed by 3 mL of 1× PBS three times. In the 2D model, the upper layer of a Transwell kit was seeded with M2 phenotype RAW246.7 and ADSC cells. For the 3D model, the upper layer of a Transwell kit was seeded with cell-loaded microspheres. The photos of the bottom layer were taken in 0, 7, 14, 21, and 28 h of incubation in both models. The corresponding recovery rate was calculated by comparing the recovery area in the same position using the Image J software.

### 4.10. qRT-PCR Analysis

The induced cells (not loaded in microspheres) were harvested and washed with 5 mL of 1 × PBS for three times. A total of 600 μL of Trizol working solution was added, and cell lysis was conducted at 4 °C in dark for 5 min. The samples were stored at −80 °C. Cell-loaded microspheres were collected and transferred into an EP tube and thoroughly grinded for 2 min. A total of 1 mL of Trizol working solution was added, and cell lysis was conducted at 4 °C in dark for 5 min. The samples were stored at −80 °C. The qRT-PCR was conducted following the standard protocol using β-actin and GAPDH as a reference, respectively [38]. The sequences of primers are shown in Appendix A.

### 4.11. ELISA Analysis

A total of 100 μL of coating liquid was mixed with a primary antibody solution and placed in an ELISA testing plate followed by incubation at 4 °C overnight. Then, the coating liquid was discarded, and the well in the ELISA plate was thoroughly washed with a buffer solution from the ELISA kit for 3 times (250 μL each time). The resulting well was subsequently injected with 200 μL diluting reagent for a 2nd coating at room temperature for 1 h. Standard and experimental samples were diluted by diluting reagent and incubated for 2 h at room temperature. Afterward, the experimental well was washed with a buffer solution 3 times, and 100 μL of testing antibody was added. The resulting sample was subsequently incubated at room temperature for another 1 h. A total of 100 μL of Avidin-HRP solution was added to the experimental well, and the sample was incubated at room temperature for 30 min. A total of 100 μL of TMB solution was added followed by incubation at room temperature for 15 min. A total of 100 μL of stopping solution was finally added to the optical density at 450 nm, and 570 nm was recorded at a plate reader to calculate the protein concentration and amount.

### 4.12. Statistical Analysis

GraphPadPrism9 and SPSS software were used to analyze the statistical data. Experimental results are expressed as mean ± SEM (standard error of the mean). When *p* < 0.05, the difference was considered statistically significant. **** means *p* < 0.0001, *** means *p* < 0.001, and ** means *p* < 0.05.

## 5. Conclusions

In general, we have constructed a 3D paracrine system by encapsulating both RAW246.7 and ADSC cells into a gelatin/sodium alginate-based hydrogel microsphere-targeting wound healing treatment. The encapsulation did not obviously affect the cell viability within 10 days of incubation. This suggested an acceptable biocompatibility of the synthesized microsphere. The qRT-PCR and ELISA analysis results showed efficient downregulation of the pro-inflammatory cytokine (IL-6) and upregulation of anti-inflammatory cytokines (Arg-1, CD206, IL-10, IL-8, bFGF, HGF, and VEGF) in the 3D paracrine system compared to that of the 2D paracrine system. Furthermore, the wound healing rate in the in vitro model was much faster upon application of the 3D paracrine system compared to that of the 2D system. The above results indicated the current 3D paracrine system might be a robust dressing material for efficient wound healing.

## Figures and Tables

**Figure 1 ijms-24-01240-f001:**
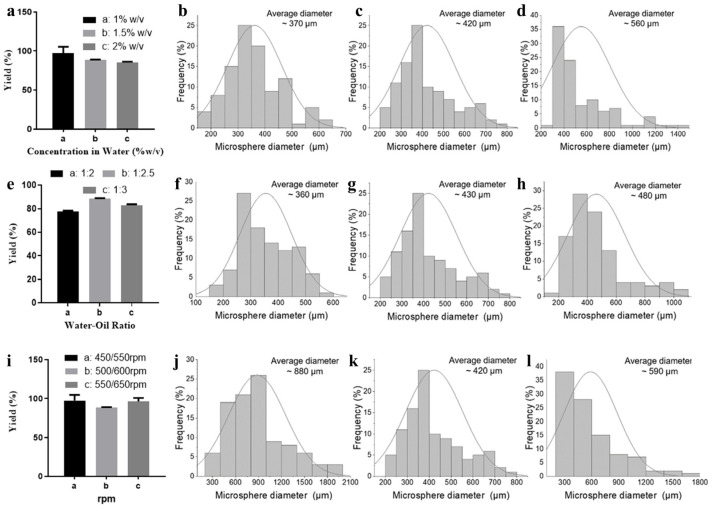
Microsphere yield, average diameter, and size distribution under different conditions: (**a**) Microsphere yield under a water–oil ratio of 1:2.5 and a stirring rate at 500–600 rpm (gelatin/sodium alginate concentration of 1, 1.5, and 2%); (**b**–**d**) average diameter and size distribution of a microsphere obtained under gelatin/sodium alginate concentration of 1, 1.5, and 2% (*w/v*); (**e**) microsphere yield under a gelatin/sodium alginate concentration of 1.5% (*w/v*) and a stirring rate at 500–600 rpm (water-oil ratio of 1:2, 1:2.5, and 1:3); (**f**–**h**) average diameter and size distribution of a microsphere obtained under a water–oil ratio of 1:2, 1:2.5, and 1:3; (**i**) microsphere yield under a gelatin/sodium alginate concentration of 1.5% (*w/v*) and a water–oil ratio of 1:2.5 (stirring rate at 450–550, 500–600, and 550–650 rpm); (**j**–**l**) average diameter and size distribution of a microsphere obtained under a water–oil ratio of a stirring rate at 450–550, 500–600, and 550–650 rpm.

**Figure 2 ijms-24-01240-f002:**
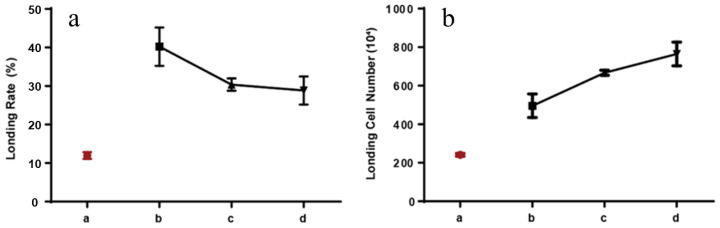
Loading rate (**a**) and loading number (**b**) of HSF and RAW246.7 cells in a microsphere: (**a**) red point was a HSF loading rate, and black points were the loading rate of RAW246.7 cells (density of 0.50 × 10^7^/mL—point b in figure (**a**); density of 1.0 × 10^7^/mL—point c in figure (**a**); density of 1.50 × 10^7^/mL—point c in figure (**a**); (**b**) red point was HSF loading number, and black points were the loading number of RAW246.7 cells (density of 0.50 × 10^7^/mL—point b in figure (**a)**; density of 1.0 × 10^7^/mL—point c in figure (**a)**; density of 1.50 × 10^7^/mL—point c in figure (**a**).

**Figure 3 ijms-24-01240-f003:**
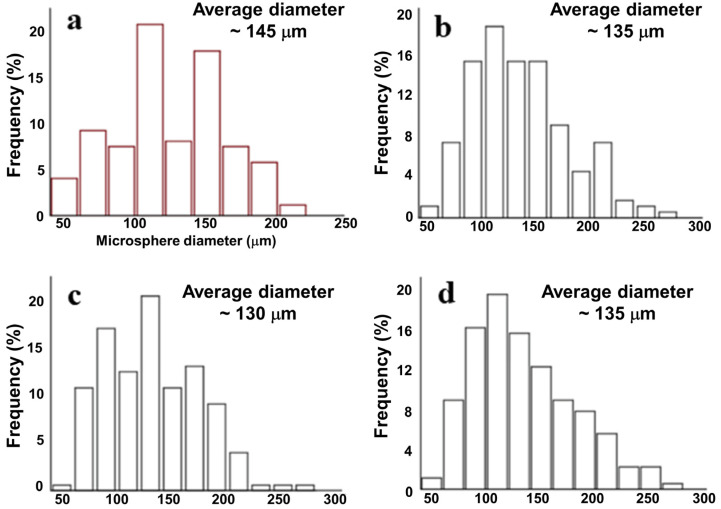
An average diameter and size distribution of cell encapsulated microspheres: (**a**) A HSF cell-encapsulated microsphere (density of 0.50 × 10^7^/mL); (**b**) RAW246.7 cell-encapsulated microsphere (density of 0.50 × 10^7^/mL); (**c**) RAW246.7 cell-encapsulated microsphere (density of 1.00 × 10^7^/mL); (**d**) RAW246.7 cell-encapsulated microsphere (density of 1.50 × 10^7^/mL).

**Figure 4 ijms-24-01240-f004:**
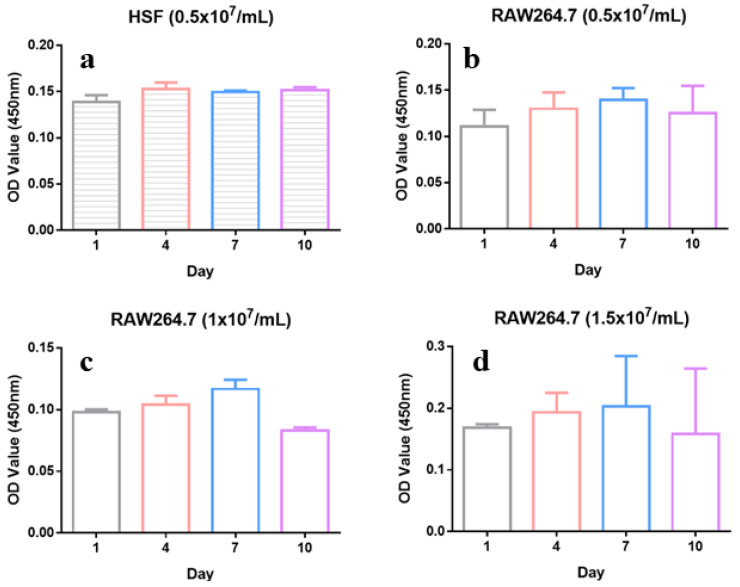
Cell viability of encapsulated HSF and RAW246.7 cells on different days: (**a**) HSF cells encapsulated at the density of 0.50 × 10^7^/mL; (**b**) RAW246.7 cells encapsulated at the density of 0.50 × 10^7^/mL; (**c**) RAW246.7 cells encapsulated at the density of 1.00 × 10^7^/mL; (**d**) RAW246.7 cells encapsulated at the density of 1.50 × 10^7^/mL.

**Figure 5 ijms-24-01240-f005:**
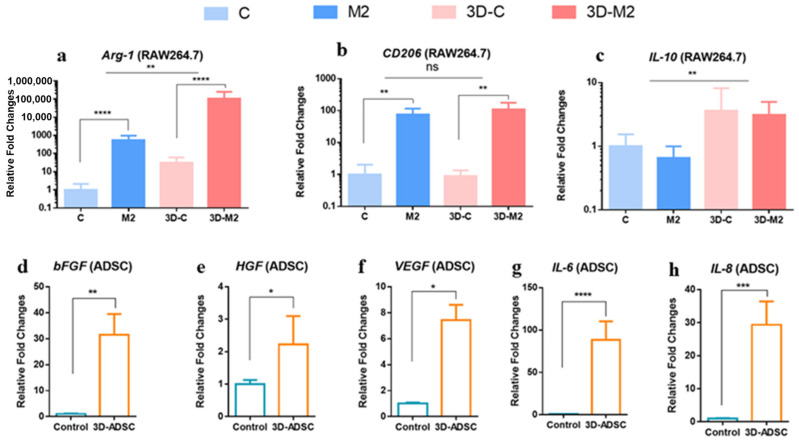
qRT-PCR results of classical cytokines expression of RAW246.7 and ADSC cells encapsulated in a microsphere and a 2D culture environment: (**a**) *Arg-1* expression from RAW246.7; (**b**) *CD206* expression from RAW246.7; (**c**) *IL-101,000* expression from RAW246.7; (**d**) *bFGF* expression from ADSC; (**e**) *HGF* expression from ADSC; (**f**) *VEGF* expression from ADSC; (**g**) *IL-6* expression from ADSC; (**h**) *IL-8* expression from ADSC. *n* = 3, * *p* < 0.05, ** *p* < 0.01, *** *p* < 0.001, **** *p* < 0.0001.

**Figure 6 ijms-24-01240-f006:**
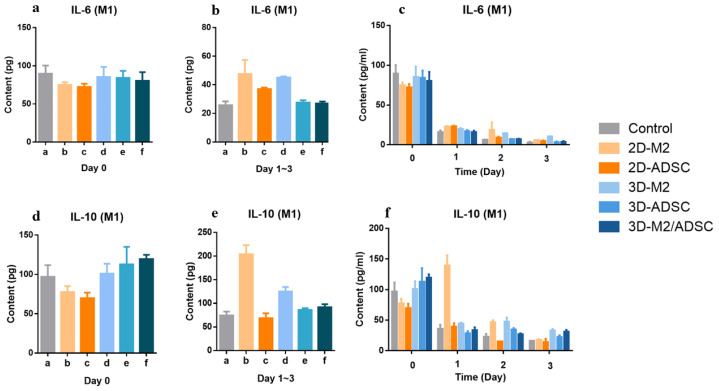
Inflammatory regulation cytokine (IL-6 and IL-10) expression level in the in vitro wound model determined by the ELISA method: (**a**) IL-6 expression on day 0 from 2D and 3D paracrine systems; (**b**) IL-6 expression from day 1 to day 3 from 2D and 3D paracrine systems; (**c**) concentration changes in IL-6 from day 0 to day 3; (**d**) IL-10 expression on day 0 from 2D and 3D paracrine systems; (**e**) IL-10 expression from day 1 to day 3 from 2D and 3D paracrine systems; (**f**) concentration changes in IL-10 from day 0 to day 3.

**Figure 7 ijms-24-01240-f007:**
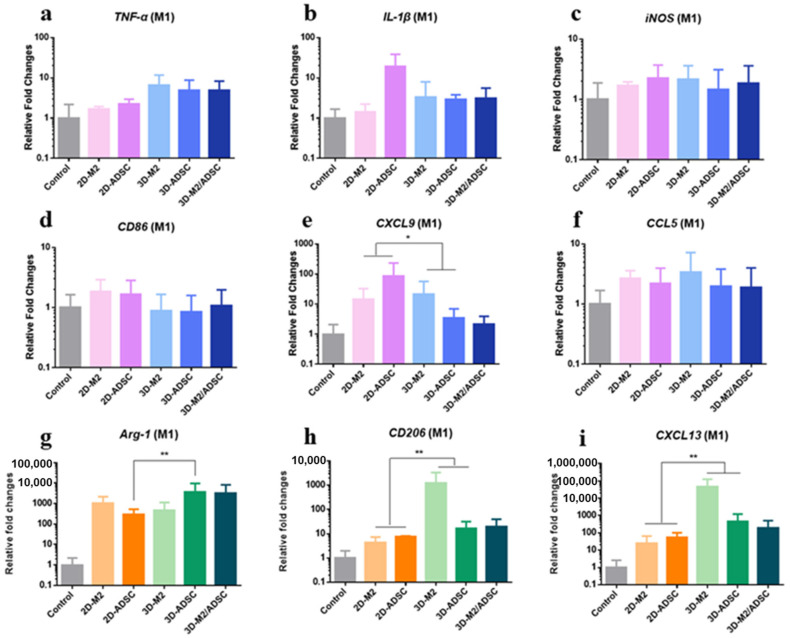
The qRT-PCR results of a classic cytokines expression from encapsulated RAW246.7 and ADSC cells into an in vitro wound model microenvironment: (**a**) *TNF-α* expression; (**b**) *IL-1β* expression; (**c**) *iNOS* expression; (**d**) *CD66* expression;(**e**) *CXCL9* expression; (**f**) *CCL-9* expression; (**g**) *Arg-1* expression; (**h**) *CD206* expression; (**i**) *CXCL13* expression. *n* = 3, * *p* < 0.05, ** *p* < 0.01.

**Figure 8 ijms-24-01240-f008:**
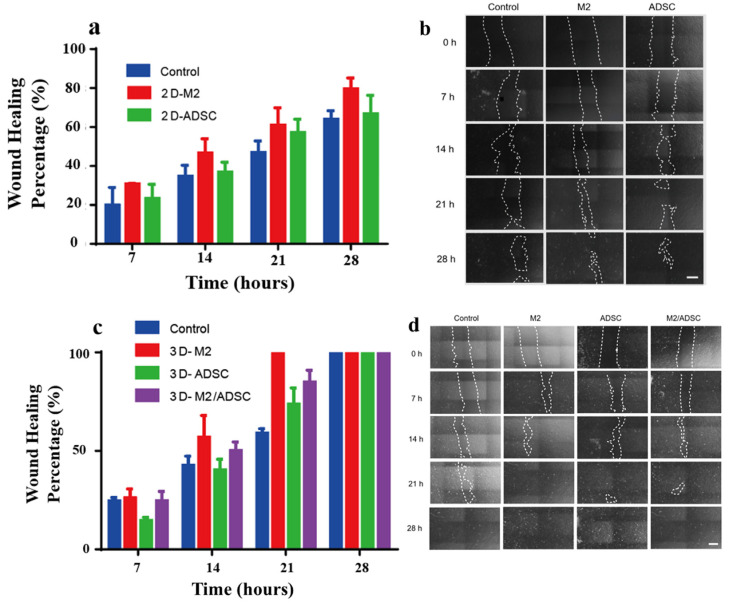
Wound healing percentage over time under the application of 2D and 3D paracrine systems in the in vitro model (**a**,**c**) and the corresponding HSF cell migrating photos (**b**,**d**).

## Data Availability

Not applicable.

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
