# Peer review of "Fabrication and Performance Evaluation of Gelatin/Sodium Alginate Hydrogel-Based Macrophage and MSC Cell-Encapsulated Paracrine System with Potential Application in Wound Healing"

_ijms, 2023, doi:10.3390/ijms24021240_

Round 1
Reviewer 1 Report
The authors embedded live macrophages and adipose mesenchymal stem cells in gelatin and sodium alginate hydrogel to construct a novel 3D paracrine system for wound healing. According to their preliminary results, this paracrine system possibility exhibited promising effects in wound healing in the aspect of positive tuning of anti-inflammatory factors together with accelerated recovery rate in 3D wound model. In my opinion, the authors proposed an interesting and new strategy in wound healing. And the manuscript should be published in the journal of International Journal of Molecular Sciences after addressing the below issues.
1. The “a, b, c, d” in the figures are suggested to be uniformly placed in the upper left corner of the Figures. The type of the characters in all the figures are suggested to be unified for a better presentation.
2. In abstract, line 3, there are two successive word “cells” in one sentence.
3. The authors should give full name in prior to the abbreviation in the first time in the manuscript. For instance, gene Arg-1 and IL-10 on page 6.
4. The authors should pay attention to the writing of chemical compositing.
5. I am not clear about Figure 6, in details, what is the difference between Figure 6a, 6b and 6c; 6d, 6e and 6f. The authors should make it more clear by recompositing the corresponding description.
6. Figure 3 should be replaced by better resolution ones. Figure 8b and 8d were too small to be clearly seen.
7. Some recent references should be cited, such as Chinese Chemical Letters 2022, 33, 5030-5034; Chinese Chemical Letters 2022, 33, 2975-2981; Engineering Reports 2020, 2, e1214; Materials Today Bio 2022, 16, 100429; Chinese Chemical Letters. 2022, 33, 1880-1884.
Reviewer 2 Report
The manuscript entitled as ‘Fabrication and Performance Evaluation of Gelatin/Sodium Alginate Hydrogel Based Macrophage and MSC Cell Encapsulated Paracrine System for Wound Healing’ seems under the scope of this journal. This study investigates the effects of Macrophages and MSC Cells on wound healing in 2D cell culture versus 3D cell encapsulated hydrogel systems based on alginate and gelatine. The studies of diverse stimuli and gene expression indicate a positive impact on wound healing in the 3D system. This study seems to have contributions to the field.
The recommendations are addressed below:
1. Introduction section can be improved by addressing similar cell-loaded hydrogel systems for wound healing in the literature. The novelty of this study should be highlighted in the last part of the introduction section.
2. Discussion section should be improved by comparing the results of this study with other related literature studies.
3. The current version of the manuscript needs to be checked grammatically since there are some incomplete sentences. Some sentences are required to be rewritten since they are not very clear.
Please especially check the below sentences:
No obviously………… at Line 17
Biomaterial and synthetic material-based dressing………… at Line 31
In this aspect, functions of macrophages centered in………… at Line 49
The sentences in Lines 56-59
The sentences in Lines 65-67
The sentences in Lines 75-76
The sentences in Lines 85-87
The sentences on Lines 95-99
The sentences on Lines 194-196
The sentences in Lines 206-207
The sentences in Lines 215-218
Remove the symbols similar to the sign of @ at Line 233-235
Check the sentences on Lines 263-265
Check the incomplete sentence between Lines 305-306
Remove repeating phrases at Lines 352-353
4. Write numbers in chemical compounds as subunits, such as CO2 instead of CO2
5. Please check the sentences starting with ‘And’, and change those with suitable phrases, such as ‘In addition, Moreover, etc.’
6. The titles given as 2.5 and 2.6 seem the same, (‘Influence on expression of Macrophage 2 phenotype and anti-inflammatory genes in encapsulated RAW246.7 and ADSC cells’) please check those.
7. When you mention the details of supplied materials or used devices, please be constant at adding the city and country information of the manufacturer/supplier in the parenthesis.
8. At line 100, ‘Figure S1’ was cited but it is not present in the manuscript, it is in Supplementary materials. Therefore, please add information in parenthesis: e.g. Figure S1 (in Supplementary materials). Please apply this for Figure S2 and others too.
9. It is beneficial to briefly explain the mentioned ‘ELISA test for inflammatory regulation protein expression analysis by standard protocol (Avidin-HRP-TMB method).’
Reviewer 3 Report
The work by Hui Chong et al. reported an interesting hydrogel cell scaffold for wound healing. Generally, the work is new and the data are solid to support their statements. I recommended that this paper can be accepted for publication in International Journal of Molecular Sciences after minor revision.
1. Title: the hydrogel material designed in the article has the potential to be used as a wound dressing, would it be more appropriate and accurate to add a potential?
2. These biolpolymers (sodium alginate and gelatin) added by the authors in the material design and their respective advantages should be explained. In this way, the reader will have a clearer idea of the features of this article.
3. The English of manuscript requires some improvements. The authors should improve language of the manuscript carefully to minimize grammatical and bibliographic errors.
4. Advantages for protein/polysaccharide dressings can be strengthened by citing 10.1016/j.mtadv.2022.100271 and what are the advantages of the current work compared to published articles?
5. It could be better if a brief comment (challenges and future prospects) is added at the manuscript. Where are these hydrogels going to be used in real life? The novelty of this work can be described at the end of the Introduction.
6. There are some formatting errors in the article. For example, spelling of references must be checked to meet the journal style (such as Reference 11). Please check carefully and use it properly.
